# Method validation and measurement uncertainty estimation of pesticide residues in Okra by GC/HPLC

**Anjana Srivastava** *, **Shishir Tandon, Gajanpal Singh, Shruti Pathak**

Department of Chemistry, Govind Ballabh Pant University of Agriculture and Technology, Pantnagar, Udham Singh Nagar, Uttarakhand, India

* anj612003@gmail.com

## Abstract

Reliability and accuracy of an analytical method is ensured by method validation technique. The present study was aimed to optimize and validate a rapid, reliable and accurate method for quantitatively determining pesticide residues of a diverse group in okra matrix. All method performance characteristics pertaining to method validation was tested. Three different pesticides viz. Thiamethoxam, Ethion, and lambda Cyhalothrin of diverse chemical classes which are applied on okra cultivation and have high MRLs as per FSSAI, were selected. Okra available in local market is often laced with these pesticides. The higher concentrations of pesticide residues in okra can be severely toxic to consumers. Thus validation of method that is simple and cost effective and can give accurate results is desirable for monitoring of these pesticides in okra. Hence a method was validated for analysis of Thiamethoxam, Ethion, and lambda Cyhalothrinby HPLC/GC. Pesticide residues fromokra samples were extracted using modifiedQuEChERs method, followed by injection into GC/HPLC. The validated method demonstrated suitable specificity, linearity, recovery etc.The calibration curves were linear for all the threepesticides with a regression coefficient, $r^2 > 0.99$. Matrix effect observed for all three pesticides in okra, fell within the range of ±20%. All pesticides were quantified successfully at a concentration of 0.30 mg/kg with an average recovery of more than 70% and a relative standard deviation (RSD) of less than 20%. The procedure was simple, rapid, cost effective and depicted high accuracy. The greenness of the method evaluated on Agro Eco Scale was satisfactory. Theestimation of uncertainties based on the validation data, werefound to be below the default limit of 50%. The quality control (QC) charts based on the basis of intra-laboratory performance were prepared at LOQ of pesticides to ensure the validity and accuracy of laboratory test results.

**Data availability statement:** All relevant data are within the paper and its Supporting Information files.

**Funding:** The authors are thankful for the financial assistance provided by the Coordinator, All India Network Project (AINP) on Pesticide Residue Analysis, Indian Agricultural Research Institute (IARI), New Delhi for carrying out these studies.

**Competing interests:** NO authors have any competing interests.

## Introduction

Several groups of pesticides are used to control weeds, insects, microorganisms and other pests. Out of the applied pesticides many of them are responsible for contaminating the produce which upon consumption by humans or animals, may pose severe adverse health impacts [1]. Farmers commonly apply mixture of insecticides to obtain good quality and quantity of fruits and vegetables but quite often after pesticides application, their residues contaminate these crops. Okra [*Abelmoschus esculentus* (L.) Moench] commonly known as lady finger, belongs to Malvaceae family and is one of the important vegetable crops of India. A high percent of shoot and fruit infestation has been reported on okra due to seasonal incidence [2].Three insecticides viz. thiamethoxam, ethion and lambda cyhalothrin which are applied on okra crop under subtropical conditions in this region and have high MRL values in okra as per Food Safety and Standards Authority India (FSSAI) [3] were chosen for performing method validation studies. Real okra samples are often laced with other pesticides too besides thiamethoxam, ethion and lambda cyhalothrin. Since they were causing interference in method validation process very few real okra samples were tested. However, regular monitoring of pesticide residues in food using accurate, reliable and low-cost analytical methods is desirable to ensure food safety [4]. For this purpose sensitive and reliable methods need to be validated in different crops. The main validation parameters for assuring reliability of the method include evaluation of precision, bias, linearity, detection limit, quantification limit, robustness, matrix effect and finally the uncertainty [5]. Uncertainty estimation for the validated method has also become one of the main focuses of interest as it confirms the data quality [16] and demonstrates the suitability of the analytical method [6]. In recent years QuEChERS method, developed byAnastassiades et al. (2003) [7], has become a common technique for multipesticide residue analysis due to its applicability to a widevariety of pesticides [8].For pesticide residue analysis, HPLC and GC are well-established, cost effective separation techniques which are employed though many improvements like development of new stationary phases and chromatographic support are incorporated in these techniques from time to time [9].

Measurement of uncertainty for the validated method to quantitatively comply with ISO/IEC 17025 requirements has also becomean essentialcriterion for method validation. Uncertainty of the measurement value (MU) parameter actually covers all the effects of operating analytical procedure that are followed during the validation process. Ultimately a single uncertainty value that has been derived from the entire measurement procedure is obtained [10]. Besides the above, preparation and interpretation of quality control (QC) charts plotted on the basis of periodic inspection of the results are also being used to identify sources of variation in the method validation process.In quality control charts, acceptance criteria are determined based on statistical data to establish upper control limits (UCL) and lower control limits (LCL). When the process is stable, sample values are likely to fall within these defined limits [11]. If any values exceed the limits, the process is regarded as unstable.In the present research work rapid method validation and measurement uncertainty estimation for determination of three multiclass pesticide residues in okra has been

developed. The studied insecticides (Thiamethoxam, Ethion, and lambda Cyhalothrin) depicted in Fig 1, are representative of three most commonly used classes, viz. neonicotinoids, organophosphates and pyrethroids, which are generally applied on okra crop of this region. During validation, different performance criteria were examined using okra crop to confirm the fitness of the results with the pre-defned criteria. The uncertainty measurements were estimated and QC charts from recovery data at LOQ were also made to ensure the validity of the results.The greenness assessment of the analytical method was also done through Analytical Eco Scale on the basis of green analytical chemistry metrics proposed by Yin et al., 2024 [12],

## Materials and methods

Three individual pesticides viz. Ethion, lambda Cyhalothrin and Thiamethoxam were procured from Dr. Ehrenstorfer, GmbH, Germany. Other reagents like HPLC grade acetonitrile, methanol, hexane, distilled water (HPLC grade) were procured from M/s Merck, India. Analytical grade anhydrous magnesium sulfate and primary secondary amine (PSA) were purchased from M/s Merck/ Thermofisher, India.

Individual stock solution of each pesticide was prepared at a concentration of 100 mg/kg by dissolving in adesired solvent, i.e., acetonitrile/ hexane. All the standard solutions were kept in a refrigerator at 4° C till use. Working solutions were made by diluting the stock solution with the appropriate solventbut before dilutions, solutions were given time to attain room temperature.

### Procurement of okra, extraction and analysis

The okra samples without any previous history of pesticide usage were collected from Vegetable Research Centre (VRC), GB Pant University of Agriculture and Technology, Pantnagar, Udham Singh Nagar, Uttarakhand, India for method validation purpose. The extraction of all the pesticides from okra was done using the modified QuEChERS extraction protocol published earlier by Srivastava et. al [13].The whole process of extraction and analysis was carried out using good agricultural practices (GAP) and mild solvents which were environment friendly. For the extraction of thiamethoxam from okra, 10g okra was taken in 50 ml centrifuge tubes in replicates. To this 10 mL of acetonitrile ($CH_3CN$) was added, followed by vortexing for 1−2 min. Next, 4 g of $MgSO_4$ and 1 g of NaCl were added, and the centrifuge tube was vortexed again for a minute.The contents of the tubes were centrifuged at 5000 rpm for 5 min. that led to the separation of organic layer. An aliquot of 1 mL from the top layer was transferred to a 10 mL centrifuge tube that contained previously weighed 150 mg of PSA and 1 g of $MgSO_4$.The tubes were centrifuged again for 1 min. at 4000 rpm. The aliquot was then passed through a 0.22 μm Polytetrafluoroethylene (PTFE) membrane disc filter, and the final extract was transferred to a sampling vial for analysis. For extraction of ethion and lambda cyhalothrin from okra, n-hexane was taken in place of $CH_3CN$ and the remaining process was similar as discussed above. The instrumentation analysis of ethion and lambda Cyhalothrin was done using GC(Thermofisher Scientific, Trace 1110) mounted with ECD and a capillary column (30 m x 0.25 mm i.d. having a film thickness of 0.25μm). The GC conditions were optimized by varying the column, injector and detector temperatures, gas flow rates etc. An initial column temperature of 100°C—increase @ 25°C min⁻¹ for 4 min. up to 180°C–increase @ 5°C min⁻¹ for 18 min. up to 270°C followed by a final ramp rate of 10°C min⁻¹ to reach to a temperature of 300°C was finalized for the pesticides analysis. The injection volume was 1 μl and the injector and detector temperatures were 250° and 300°C, respectively. Nitrogen (99.99% purity) was used as the carrier gas at a flow rate of 1.2 ml min⁻¹ and the total run time was taken as 28.5 min.for the elution of the two pesticides. Under the abovementioned GC-ECD conditions the peaks of the pesticides were well resolved and retention times of ethion and lamda Cyhalothrin were 17.08 and 21.48 min. respectively. Analysis of Thiamethoxam was achieved using Dionex Ultimate 3000 HPLC system, equipped with Thermofisher RP-C18 column (250x4.6mm i.d.) (particle size-5 μm), injector loop of 20μl, UV-VIS detector and dual pump. The optimum HPLC conditions were mobile phase: $CH_3CN$-$H_2O$ (60:40), UV detection at 254 nm wavelength and flow rate: 0.5 ml min⁻¹. The retention time ($t_r$) of Thiamethoxam was found to be 5.81 min.

**Thiamethoxam**

**Ethion**

**λ- Cyhalothrin**

**Fig 1. Chemical structures of insecticides.**

## Validation experiments

The method validation of all the three pesticides was done as per European Commission document, SANTE guidelines 2021 [14] with special interest to specificity, linearity, matrix effects, limits of detection (LOD), limits of quantification (LOQ), accuracy and precision, recovery, robustness and estimation of measurement uncertainty.

## Specificity/ Selectivity

The specificity/ selectivity of the experiment was assessed by comparing the chromatograms of blank, standard, and sample solutions, each in three replicates.

## Linearity and matrix match

The linearity of the method was determined by preparing a standard stock solution of 100 mg/kg of each of the pesticide and then diluting each of them to 10 mg/kg. Further dilutions of all the pesticides to six lower concentrations ranging from 0.05 to 2 mg/kgfor ethion and lambda cyhalothrin and 0.1 to 5 mg/kg for thiamethoxam were done using hexane/ acetonitrile solvent. For matrix match the extract of matrix sample was taken and fortified with six dilutions of the pesticides as done above.

A calibration curve for the pesticides for both the solvent and matrix match were plotted as peak area vs. concentration (mg/kg). Matrix effects was evaluated by using the standard formula (i) on the basis of measured relative peak areas of calibration standards in solvent and the areas in the relevant matrix.

$$\% \text{ Matrix interference} = \frac{\text{Peak area of pesticide in solvent} - \text{Peak area of pesticide in matrix}}{\text{Peak area of pesticide in solvent}} \times 100$$

## LOD and LOQ

The LOD and LOQ values were determined with the help of the calibration curve and regression equation of the linearity graph using the mathematical equations

$$\text{LOD} = 3.3 \text{ x } \sigma/\text{S and LOQ} = 10 \text{ x } \sigma/\text{S}$$

Where σ = Standard deviation of the intercept and S = Slope of calibration curve.

## Recovery, accuracy and precision

The recovery experiments were conducted at three different concentrations (LOQ, 5LOQ and 10 LOQ) in five replicates of each. Extraction and clean-up was done as per the QuEChERS method described above. Recovery percent was calculated using the given formula. Standard deviation (SD) and % Relative standard deviation (RSD) were calculated

## Recovered concentration

$$\% \text{ Recovery} = \frac{\text{Recovered concentration}}{\text{Spiked concentration}} \times 100$$

The accuracy of the developed method was established using the data of recovery studies, SD and % RSD values.

For precision studies repeatability tests were performed by injecting ten replicate samples of all the three targeted pesticides at one test concentration.

## Robustness

Robustness of the method was determined by making deliberate changes in analytical instrumentation conditions. For GC, robustness was measured by checking for variations in the carrier gas flow rate, and on the initial oven programming temperature whereas for HPLC method, robustness was determined by making changes in the mobile phase ratio and detection wavelength of the pesticide. The differences in the responses of pesticides were recorded and % RSD was calculated. For each set of variation, five replicate injections of the standard solution were done.

## Uncertainty measurement

Uncertainty was estimated by the standard deviation calculated on within laboratory reproducibility. Measurement uncertainty values were estimated using a top-down approach based on the validation data [15,16]. The expanded uncertainty (Uc) at a confidence level of 95% was obtained using the formula:

$$Uc = \sqrt{\{(u1)^2 + (u2)^2 + (u3)^2 + (u4)^2 + (u5)^2 + (u6)^2 + (u7)^2 + (u8) + (u9)\}}$$

where u1 – u7 are different sources of uncertainty. These include uncertainties ofrepeatability (u1); purity of standard CRM(u2); weighing balances (u3, u4); volumetric flask (u5); micropipettes (u6, u7); recovery (u8) and linearity (u9). For standard purity $u$ ($P$), the standard deviation (SD) of 0.02 and for rectangular distribution ($d = 3$) was considered.The approach is based on precision data generated during method validation.

## Preparation of Quality Control (QC) charts

QC chart for every pesticide were recorded at LOQ recovery level through intra-laboratory performance in terms of the mean and of the ±3 standard deviations (sdv) confidence band [10].

## Greenness assessment

For assessing the greenness of the validated procedure Analytical Eco Scale (AES) evaluation was performed taking into consideration the penalty points (PPs) caused due to the parameters like weight of the substances used, toxicity of solvents or chemicals used in the study, temperature during the validation process and the instruments employed in the validation process.

## Results and Discussion

### Validation results

**Specificity/ selectivity.** The specificity/ selectivityin method validation is the ability of a method to measure a target analyte without interference from other components in a sample.Specificity of the method based on the chromatographic peak purity was observed in the chromatograms [17]. There was no interference of any other matrix peak that would impede the analysis of the pesticide peak of interest in either of the three insecticides as depicted in overlaid chromatograms of thiamethoxam, ethion and lambda Cyhalothrin (Fig 2).

**Linearity.** The linearity of the method was established by plotting a graph between mean peak area and concentration. Linearity of calibration curves for all the three pesticides was evaluated using linear regression analysis and. co-relation coefficient value and has been depicted in Table 1. As evident from the table, linear correlations were obtained between absorbance and concentration with high $R^2$ values confirming that the analytical method validation met the acceptance criteria [16] both for the solvent as well as matrix match. Hence linearity of method was proved over the concentration range of 0.05–2 mg/ Kg for ethion and lambda cyhalothrin and between 0.1–5.0 mg/ kg for thiamethoxam. Similar linear combinations of these pesticides analysed by HPLC/GC in tomatoes, cabbage heads and cucumbers have also been reported [18–21].

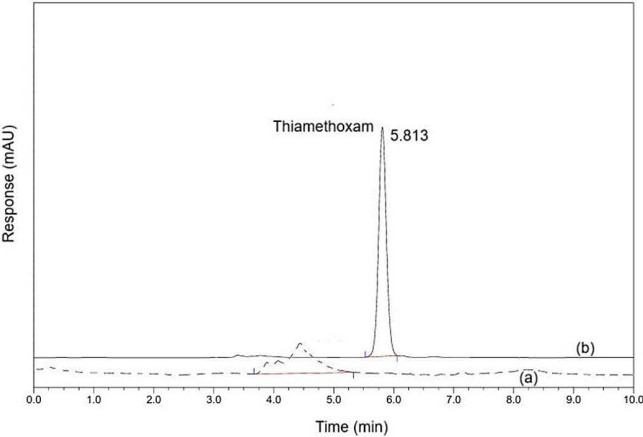

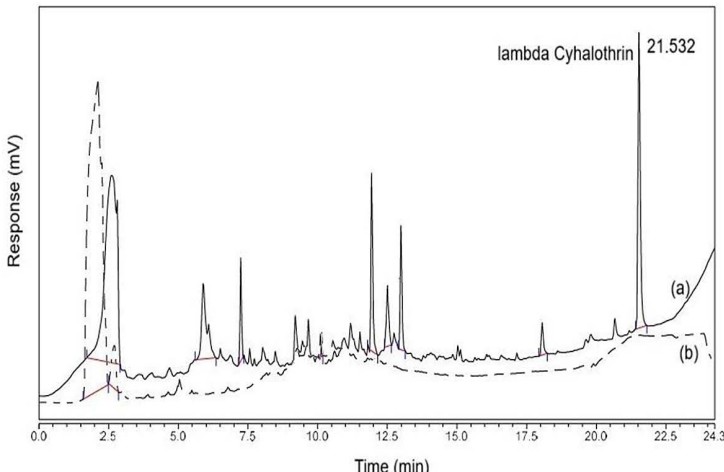

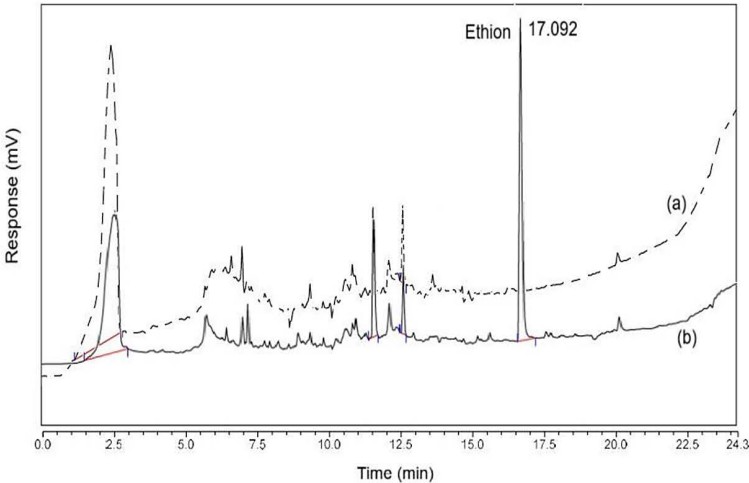

**Fig 2. Overlaid chromatograms of (a) okra blank and (b) matrix matched standard solution of Thiamethoxam Ethion and lambda Cyhalothrin.**

**Table 1. Linearity parameters of the pesticides in solvent and matrix (okra).**

| Pesticide | Calibration range (mg/kg) | Regression equation | R² | LOD (mg/kg) | LOQ (mg/kg) |
|---|---|---|---|---|---|
| Ethion in hexane | 0.05–2.0 | y = 23.36x + 0.182 | 0.998 | 0.092 | 0.277 |
| Ethion in matrix | 0.05–2.0 | y = 24.11x + 0.628 | 0.998 | | |
| Lambda cyhalothrin in hexane | 0.05–2.0 | y = 12.00x − 0.143 | 0.999 | 0.091 | 0.275 |
| Lambda cyhalothrin in matrix | 0.05–2.0 | y = 12.55x − 0.048 | 0.999 | | |
| Thiamethoxam in acetonitrile | 0.1–5.0 | y = 2.287x − 0.030 | 0.999 | 0.086 | 0.259 |
| Thiamethoxam in matrix | 0.1–5.0 | y = 2.33x + 0.017 | 0.999 | | |

The LOD and LOQ values (Table 1) calculated from the data of calibration curve using the mathematical formula ranged between 0.086–0.092 for LOD and between 0.259–0.277 for LOQ.

Matrix matched recovery studies to include the interferences (if any) from the okra matrix were performed. Recovery studies using blank matrices spiked at three concentration levels (LOQ, 5 LOQ and 10 LOQ) were done by taking five replicates of each concentrationprior to sample preparation. The sample concentrations, recovery and relative standard deviation (% RSD) were calculated and depicted in Table 2. All the recovery values were in the range of 70–120% with % RSD < 20, and thereby deemed acceptable according to SANTE (2017) [22] guidelines.

The results of robustness of test method as demonstrated by change in mobile phasecomposition and absorption wavelength in detection of thiamethoxam by HPLC-UV and by variation in gas flow rate and oven temperature programming during detection of ethion and lambda Cyhalothrin by GC-ECD are depicted in Table 3.

It is evident from the data that there was only minor variation in the peak areas of the pesticides by changes in detection parameters and the % RSD in all the cases was < 20%. Thus the analytical procedureremained unaffected by minor deliberate variations in method parameters which confirm that the used method is robust. However, for analyzing ethion, lambda-cyhalothrin, and thiamethoxam in okra or other vegetables, both GC-MS and LC-MS methods are viable options through which lower LOQs can be obtained which can meet the MRLs set by EU. Several studies on method validation of pesticides in vegetables, fruits and even other commodities have been reported. [18,23]. Due to nonavailability of these expensive instruments in the lab method validation was performed using GC and HPLC and through this also satisfactory results of all the required method validation parameters were obtained.

In pesticide residue analysis, measurement uncertainty is critical during compliance statements against a standard. Measurement uncertainty (MU) values, estimated using a top-down approach are based on the trueness and precision data generated in the method validation experiment to estimate the MU value. The range of MU values indicates where the true value of a measurement is likely to be, thus reflecting the variability in the measurement process.

MU values were calculated using intra-laboratory validation data and the combined expanded uncertainty was calculated by multiplying the MU obtained by a coverage factor (k) of 2 using the formula for expressing it at 95% confidence level.

$$Uc = k \times u$$

**Table 2. Recovery data of pesticides at three concentrations from okra matrix.**

| Pesticide | % Recovery at LOQ | % RSD | % Recovery at 5 LOQ | % RSD | % Recovery at 10 LOQ | % RSD |
|---|---|---|---|---|---|---|
| Ethion | 81.39 | 2.71 | 86.84 | 3.24 | 87.55 | 2.25 |
| lambda Cyhalothrin | 81.91 | 2.62 | 88.85 | 2.33 | 94.04 | 2.45 |
| Thiamethoxam | 77.37 | 2.5 | 86.04 | 2.04 | 90.19 | 2.24 |

**Table 3. Robustness data of Ethion, Lambda Cyhalothrin and Thiamethoxam.**

| Pesticide | Detection method | Parameter | Peak area (mean of five replicates% RSD | | | |
|---|---|---|---|---|---|---|
| Ethion | GC-ECD | Gas Flow rate (ml min⁻¹) | Original | 1.2 | 5.51 mV*min | 9.79 |
| | | | Changed | 1.5 | 5.67 mV*min | 12.61 |
| | | Oven temperature (°C) | Original | 100 | 5.51 mV*min | 9.79 |
| | | | Changed | 60 | 5.40 mV*min | 3.27 |
| lambda Cyhalothrin | GC-ECD | Gas Flow rate (ml min⁻¹) | Original | 1.2 | 2.36 mV*min | 8.74 |
| | | | Changed | 1.5 | 2.45 mV*min | 6.70 |
| | | Oven temperature (°C) | Original | 100 | 2.36 mV*min | 8.74 |
| | | | Changed | 60 | 2.56 mV*min | 2.112 |
| Thiamethoxam | HPLC-UV | Mobile phase ratio (ACN: Water) | Original | 60: 40 | 4.55 mAU*min | 0.53 |
| | | | Changed | 65: 35 | 4.23 mAU*min | 0.97 |
| | | Detector wavelength (nm) | Original | 254 | 4.55 mAU*min | 0.53 |
| | | | Changed | 260 | 4.12 mAU*min | 1.62 |

The MU value obtained for the target pesticides viz. thiamethoxam, ethion and lambda Cyhalothrin, were found to be 0.038, 0.053 and 0.029 mg/kgrespectively at 0.5 mg/kg. As depicted in Fig 3, the % values were calculated and were found to be lower than 25% which is the default value employed for nonfatty matrixes (fruit, vegetables, and grain) by many regulatory authorities for enforcement decisions [23]. Lower uncertainty values reflect that the results are closer to the true value, with less variation and doubt around the measurement and thus increase confidence in the results. Hence the obtained values can be considered suitable for method validation of the three insecticides (thiamethoxam, ethion and lambda cyhalothrin). Shrestha et al. [4] have also reported the MU of 26 pesticides in tomato confirming the MU values to be < 50% default value which is usually employed by many regulatory authorities for enforcement decisions.

Quality control (QC) charts of all the three insecticides were made at LOQ from the data obtained for recovery percent for four consecutive days to ensure that the data is accurate, as it is necessary for valid conclusions regarding consumer exposure to pesticides and compliance with maximum pesticide residue limits. The QC charts allow a detailed knowledge

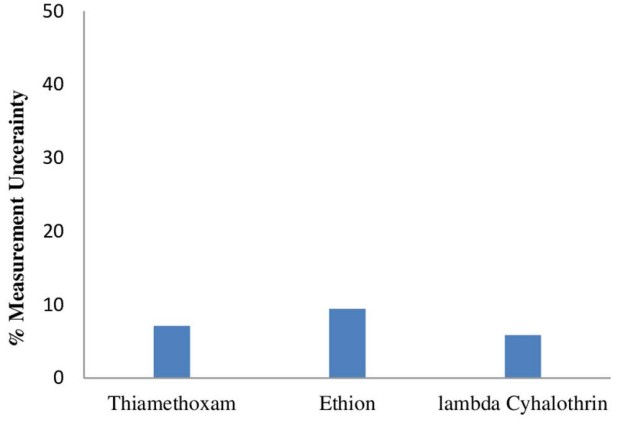

**Fig 3. Measurement uncertainty (%) for pesticides expressed at a 95% confidence level.**

on the whole palette of pesticides that are analysed and also on the changes in the course of time. Fulling [24] also performed studies on quality control (QC) chart for determination of soil organic carbon by instrumental analysis. They concluded that QC chart method is straightforward to use, easy to learn, and can quickly identify any outliers or unusual patterns in the data. Thus they make the data highly valuable for test analysis and help to ensure the precision of laboratory test outcomes.The intra-laboratory performance, represented by mean values along with the±2 and ±3 standard deviation (SD) confidence bands, is illustrated in QC charts in Fig 4a–4c. The chart includes both the upper control limits (UCL) and lower control limits (LCL).Mean (μ) and sigma (σ) values were derived from the validation data. In the figure, the limit lines indicate the mean plus or minus 1, 2, and 3 SD. Under the assumption of a Gaussian or normal distribution, approximately 68% of the data points are expected to reside within 1 SD of the mean, 95% within 2 SD of the mean, and 99.7% within 3 SD of the mean.

Typically, the mean recovery remained around 100% in case of all the insecticides. Fuling et al. [24] in their studies have also confirmed that if the setting point in QC chart is inside the control lines, it proves that the results are reliable. If it

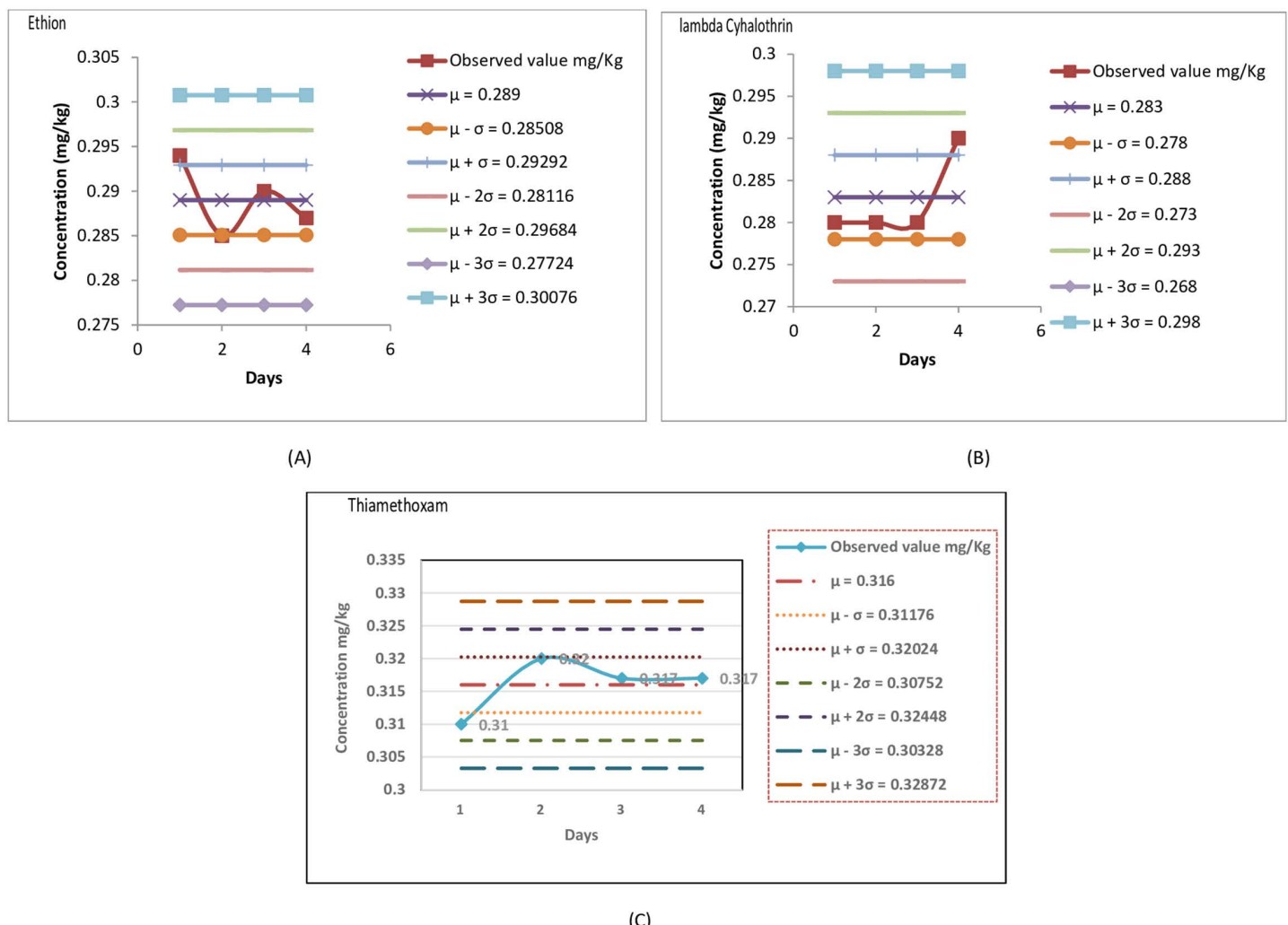

**Fig 4. Quality Control charts of (A) Ethion (B) lambda Cyhalothrin and (C) Thiamethoxam at LOQ.**

is outside the upper and lower control lines, then the test results have deviations. All the spikes were within 2SD and thus the results can be considered to be in good category. Since all the setting points lie inside the control lines, it proves that the results are reliable.

The greenness assessment was done through Analytical Eco Scale (AES) evaluation for mitigating the unfavorable effects of analytical activities on human safety, human health, and environment. The penalty points (PPs) were taken into account to calculate analytical AES. The parameters covered were weight of the substances used, the hazard of solvents or chemicals, energy required on the basis of temperature employed and the instruments employed in the analytical process, It was found that the cumulative PPs difference with 100 was 65, indicating a valid green method development procedure [25].

## Conclusion

Pesticide residues in fruits and vegetables are an important matter of public concern. In this research we have optimized and validated quick and cost effective method to quantitatively determine the amount of three commonly applied diverse group of pesticide residues in okra samples. All method validation performance characteristics were found to be satisfactory within the recommended limits, indicating the reliability of the method. Realistic uncertainty estimates are important to ensure that the results are valid. The uncertainties in the present method validation procedures were well below the recommended limits. The intra-laboratory performance in terms of the mean and of the ± 3 standard deviations (sdv) confidence band displayed in the form of QC charts also ensured that the results of validation study are accurate. The greenness assessment of the analytical method was also done using Analytical Eco Scale (AES) evaluation which confirmed that the validated analytical procedure fell within the acceptable range. Food safety is very important for human health and so for identifying pesticide residues in edible commodities especially vegetables and fruits, it is essential to apply validated analytical methods for their extraction and analysis because they are considered to provide reliable results. Besides that the developed method should also be eco-friendly so that it is safe for the environment. The validated methods in the presentstudy can thus be successfully employed for determination of thiamethoxam, ethion and lambda cyhalotrhin insecticides in okra crop

## Supporting information

**S1 File. It includes all the details of the parameters used for method validation of ethion insecticide.**
(DOCX)

**S2 File. It includes all the details of the parameters used for method validation of lambda Cyhalothrin insecticide.**
(DOCX)

**S3 File. It includes all the details of the parameters used for method validation of Thiamethoxam insecticide.**
(DOCX)

## Acknowledgments

The authors are thankful to the Head Department of Chemistry and Dean College of Basic Sciences, GB Pant University of Agriculture and Technology, Pantnagar for providing necessary facilities to undertake the above study.

## Author contributions

**Conceptualization:** Anjana Srivastava.

**Formal analysis:** Gajanpal Singh.

**Funding acquisition:** Anjana Srivastava.

**Methodology:** Shishir Tandon, Shruti Pathak.

**Validation:** Gajanpal Singh.

**Visualization:** Anjana Srivastava, Gajanpal Singh.

**Writing – original draft:** Shruti Pathak.

**Writing – review & editing:** Shishir Tandon.

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
