## [Decision Letter · Decision Letter 0]

28 Apr 2025

Dear Dr. Srivastava,

Thank you for submitting your manuscript to PLOS ONE. After careful consideration, we feel that it has merit but does not fully meet PLOS ONE’s publication criteria as it currently stands. Therefore, we invite you to submit a revised version of the manuscript that addresses the points raised during the review process.

We look forward to receiving your revised manuscript.

Kind regards,

Trung Quang Nguyen

Academic Editor

PLOS ONE

https://www.e3s-conferences.org/articles/e3sconf/pdf/2020/66/e3sconf_icgec2020_01004.pdf

http://www.dsea.unipi.it/Members/balestrinow/CP/file/QC_westgard_SPC.doc

https://onlinelibrary.wiley.com/doi/10.1155/2024/3846392

https://repositorio.ufsm.br/bitstream/handle/1/20794/TES_PPGCTA_2015_SCHWANZ_THIAGO.pdf?isAllowed=y&sequence=1

In your revision ensure you cite all your sources (including your own works), and quote or rephrase any duplicated text outside the methods section. Further consideration is dependent on these concerns being addressed.

“The authors are thankful for the financial assistance provided by the Coordinator, All India Network Project (AINP) on Pesticide Residue Analysis, Indian Agricultural Research Institute (IARI), New Delhi for carrying out these studies.”

Reviewers' comments:

Reviewer's Responses to Questions

**Comments to the Author**

1. Is the manuscript technically sound, and do the data support the conclusions?

Reviewer #1: Yes

Reviewer #2: Yes

Reviewer #3: No

Reviewer #4: Yes

Reviewer #5: Yes

Reviewer #6: Partly

2. Has the statistical analysis been performed appropriately and rigorously?

Reviewer #1: Yes

Reviewer #2: Yes

Reviewer #3: I Don't Know

Reviewer #4: Yes

Reviewer #5: Yes

Reviewer #6: No

3. Have the authors made all data underlying the findings in their manuscript fully available?

Reviewer #1: No

Reviewer #2: No

Reviewer #3: No

Reviewer #4: Yes

Reviewer #5: Yes

Reviewer #6: No

4. Is the manuscript presented in an intelligible fashion and written in standard English?

Reviewer #1: Yes

Reviewer #2: No

Reviewer #3: No

Reviewer #4: Yes

Reviewer #5: Yes

Reviewer #6: No

Reviewer #1: In this manuscript, the authors present a combination of GC and LC methods to measure three pesticides in Okra. They validate both methods and estimate their uncertainty according to standard guidelines. In my opinion the work is sound, the manuscript is well-structured and all necessary information is provided to the reader so the work could be repeated. The detectors used in combination with the chromatographic devices are admittedly not the most selective, but the authors demonstrate that selectivity is acceptable, notably via figure 2. It is also true that having to apply two different methods for three analytes may be tedious in routine, while HPLC-MS/MS might be able to monitor all three analytes (among others) with higher sensitivity. This brings me to my major comment: with LOQs of about 0.25 mg/kg, these methods cannot reach the MRLs defined e.g. by the European Commission. I haven’t found specific MRLs for Okra, but MRLs generally become more and more stringent and range from 0.01-0.05 mg/kg for the molecules analyzed. I think the authors should make a comment on this. Another important point is that there was no application to real samples to evaluate the applicability of the methods and the proportion of contaminated samples on a selection of plants.

Apart from this, I only have minor comments:

Line 56 : « to control »

Line 58 : « may pose »

Line 59 : « pesticide mixtures »

Line 69 : « precision » is mentioned twice

Line 93: “neonicotinoids”

Line 113: make it clearer that you used an extraction protocol of yours which had been published earlier.

Figure 4: I cannot see the line for the observed values in the plot at 1 mg/kg.

Finally, I noticed that the quality of English is quite variable throughout the manuscript. They are several grammatical and spelling mistakes in particular in the abstract and introduction. I suggest the paper to be reviewed by a native English scientist before publication.

Reviewer #2: Detailed Review and Required Revisions for Manuscript Submission to PLOS ONE

Upon comprehensive evaluation of the manuscript titled “Method validation and Measurement uncertainty estimation of diverse group of Pesticide residues in Okra by GC/HPLC,” it has been found that the manuscript aligns with the scope and publication criteria of PLOS ONE, especially in the domains of analytical method development, food safety, and environmental chemistry. However, to ensure clarity, consistency, and full compliance with journal standards, the following detailed observations and revisions are recommended:

Title and Abstract

• Issue: Capitalization inconsistency in the title. Avoid redundant adjectives like “rapid, reliable and accurate” unless differentiated.

• Suggested Fix: “Method Validation and Measurement Uncertainty Estimation of Pesticide Residues in Okra by GC/HPLC”

Abstract issues:

The introduction lacks context on why pesticide detection in okra is significant.

Phrasing issues: e.g., “logical specificity” is unclear.

• Revision Tip: Clarify the gap in literature and succinctly mention the novel aspect (okra matrix focus, combined GC/HPLC validation, etc.).

• Issue: The abstract lacks clear context on the problem being addressed.

• Comment: It starts directly with the validation technique but should first highlight the significance of monitoring pesticide residues in vegetables like okra.

• Suggested Revision: Introduce the relevance of pesticide residues in public health and food regulation before explaining the method.

Introduction

• Comment: Redundant mentions of pesticide contamination and vague claims (e.g., "people apply pesticides mixture").

• Scientific Gaps: Needs citation for the importance of okra in Indian agriculture and for typical pesticide usage trends.

• Suggestion: Consolidate overlapping statements and replace colloquial language with scientific phrasing.

• Strengths: It establishes a good rationale for pesticide analysis in okra, citing pest susceptibility and agricultural practices.

• Issues: Redundant mention of ISO/IEC 17025 uncertainty requirements (lines 70–80).

Phrasing like “people apply pesticides mixture” is awkward.

Use of outdated or non-specific references (e.g., Pestology 1984 [2]).

• Suggested Edits:

Consolidate mentions of measurement uncertainty.

Improve grammar and update the citation base with more recent references from 2015–2024.

Materials and Methods

• Strength: The method uses validated protocols (QuEChERS, SANTE) and details equipment clearly.

• Issue: Lack of justification for pesticide selection beyond availability.

• Detail to Add: Describe why Thiamethoxam, Ethion, and Lambda-Cyhalothrin were selected—frequency of detection, toxicity ranking, or MRL limits.

• Formatting Issue: Inconsistent units (e.g., “mg/Kg” vs. “mg/kg”), overly condensed chromatography conditions.

Validation Parameters

• Comment: Tables are well-organized and clearly demonstrate specificity, linearity, LOD, LOQ, recovery, precision, and robustness.

• Detail Gaps:

The correlation coefficients (R²) and regression equations are present but lack residual analysis.

No comparative method or matrix challenge was mentioned.

• Suggestion: Briefly compare values with standard methods in tomatoes or cucumbers to add external validity.

Measurement Uncertainty

• Strength: Comprehensive top-down approach, supported by ISO and SANTE references.

• Formatting: Equation formatting is basic; LaTeX-style or clearer breakdowns would help (e.g., clarify the meaning of “u1–u7” terms).

• Data Depth: Mention of 25% default MU limit is excellent but should elaborate on why lower values validate reliability in vegetables.

Results and Discussion

• Scientific Soundness: Recovery rates (70–120%) and RSD (<20%) meet international standards (SANCO/SANTE).

• Weaknesses:

The discussion does not critically compare GC vs. LC methods or address cross-matrix reproducibility.

Statements like “proved that the results are reliable” need substantiation or tone down.

• Suggestion: Use a more critical, peer-like tone. Acknowledge limitations and suggest future validation for other crops.

• Strengths: Presents full data on linearity, recovery, precision, and robustness. Good use of statistical treatment and visual quality control checks.

• Issues:

References like “(15)” or “(Fig. 2 (i-iii))” are not adequately contextualized. The discussion lacks depth on method comparison or implications of the findings.

• Suggestions:

Add a short comparative paragraph on how this method performs versus LC-MS/MS or published QuEChERS-GC methods.

Discuss limitations or matrix effects more critically.

Figures and Tables

• Issue: Figures referenced (e.g., Fig. 1, 2, 3, 4) are not embedded in the main text. This reduces comprehension and violates PLOS ONE’s formatting standards.

• Suggested Fix: Ensure all figures are uploaded and linked in submission; each should have a complete caption explaining what the reader should observe.

Conclusion

• Comment: Summarizes the work adequately but contains redundancy with abstract and results.

• Suggestion: Make it more outcome-oriented—emphasize real-world application in food labs and public safety monitoring.

References

• Comment: Well-curated set of primary references including SANTE, ISO 11352, and Shrestha et al.

• Issue: Duplicates found (Shrestha 2024 appears twice); fix citation style inconsistencies (some lack DOIs).

The citation style is inconsistent and lacks uniform formatting.

• Suggestion: Convert to PLOS style (Author, Year) throughout the main text and reference list. Reformat all references in line with PLOS ONE citation style (Author, Year), especially in the main text.

Questions Inferred from Manuscript

1. Could this method be extended or adapted for detecting similar residues in other vegetables or high-water content matrices like cucumber?

2. How does this method compare detection limits and cost to advanced LC-MS/MS-based multi-residue protocols?

3. Are the measurement uncertainties consistent across different analysts or labs (interlaboratory validation potential)?

Moreover, vague terminology and imprecise phrasing occasionally lead to ambiguity, as seen in descriptions of matrix effects or statistical outcomes. While the scientific merit remains intact, the overall presentation would benefit significantly from a thorough language edit by a native English speaker or a professional scientific editor. Enhancing linguistic accuracy and consistency will greatly improve the manuscript’s clarity, academic rigor, and alignment with the stylistic standards expected by journals like PLOS ONE.

Recommended for publication in PLOS ONE pending minor to moderate revisions. The study provides valuable data on pesticide residue analysis in okra using established analytical protocols. Enhancements in scientific writing, figure integration, and contextual discussion will significantly improve clarity and impact.

Reviewer #3: This manuscript presents a method validation study for the quantification of pesticide residues (Thiamethoxam, Ethion, and Lambda-Cyhalothrin) in okra using GC/HPLC. While the topic is of potential relevance for food safety and pesticide residue analysis, the current version of the manuscript suffers from several fundamental shortcomings that prevent a proper evaluation of the work and, ultimately, undermine the scientific value of the study.

Due to the critical lack of data, inadequate method description, and absence of real-world applicability, the manuscript is not suitable for publication in its current form and should be rejected. However, if the authors are able to address the concerns outlined above—particularly by including full datasets, improving the methodological rigor, and conducting additional validation experiments a revised version could be considered for re-evaluation.

Major Concerns

1. Lack of Supporting Data and Transparency in Calculation

A critical shortcoming of the manuscript is the absence of supporting data. Essential information and raw data underlying the method validation and uncertainty estimation are not provided.

The manuscript does not explain how the measurement uncertainty (MU) was calculated, nor are the input values or assumptions documented. For a method that may be used in ISO/IEC 17025-accredited laboratories, this level of detail is imperative.

A supplementary section containing raw data, calculation spreadsheets, and validation outputs would greatly enhance transparency and reproducibility.

2. Lack of Real Sample Analysis and Reference Materials

The study appears to rely solely on spiked matrixes without including contaminated real-world okra samples. This limits the applicability and relevance of the method to actual monitoring scenarios.

3. Matrix Considerations and Source Variability

The authors did not account for potential variability introduced by different varieties or sources of okra. Since agricultural matrices can vary significantly, especially in complex plant matrices like okra, it is essential to consider source-related effects on method performance.

4. Selectivity and Interferences

The manuscript lacks a proper assessment of selectivity. In the absence of mass spectrometric detection, structural analogs or co-eluting compounds may compromise the method's specificity. E. g, there is no mention of interference studies involving structurally similar pesticides that may be present in real samples.

5. Diastereomer Considerations for Lambda-Cyhalothrin

Lambda-Cyhalothrin is known to be a mixture of diastereomers that are separable by GC. The paper does not address whether these isomers were resolved in the method, and if not, how quantification was handled. This is a critical omission, as the relative proportions of isomers can influence the accuracy of the method.

6. Inadequate Method Description and Reproducibility Concerns

The QuEChERS extraction protocol is not described in sufficient detail to ensure reproducibility. The manuscript refers to a paper that references another paper, instead of clearly describing the procedure used. A self-contained, step-by-step description is necessary for validation and replication.

7. Robustness and Sample Preparation

An insufficient robustness study was conducted, that relied solemnly on the instrumental (detection) parameters. Key aspects of sample preparation (e.g., drying, grinding, extraction, cleanup) are major contributors to variability and measurement uncertainty. The impact of these factors should have been assessed both in the robustness study and in the uncertainty estimation.

Minor Comments

1. Figures and Visual Data Quality

Figure 1: The chemical structures lack uniformity in style; the structures of Ethion and Cyhalothrin appear distorted.

Figure 2B: The observed value line is missing, making it impossible to interpret the graph properly.

2. Units

The authors should use SI-compliant units such as “mg kg⁻¹” or “mg/kg,” following recommendations by IUPAC, BIPM, and other standards bodies.

3. Language and Typos

The manuscript requires language polishing. Several typographical and grammatical errors are present. For example:

Line 131: “Europeon [sic!] Commission” should be “European Commission”

Line 69: The phrase “assays of precision, bias, linearity, precision…” is redundant and unclear.

Reviewer #4: Reviewer comments:

I sincerely appreciate the editor's suggestion that I revise this manuscript.

After the first revision, I suggest that the manuscript can be accepted.

The following specific points need to be considered during the revision of the manuscript.

1) At abstract, line number 36 please modify 0.3mg/kg to be 0.30 mg/kg.

2) Please justify the alignment for the whole manuscript.

3) Please try to increase the figures resolution

4) At Introduction line 56: modify from Several groups of pesticides are used to for controlling weeds to be Several groups of pesticides are used for controlling weeds.

Reviewer #5: Comments to the Authors:

The aim of the study is well-defined: optimizing and validating a method for pesticide residue analysis in okra.

1-The selection of pesticides (Thiamethoxam, Ethion, and lambda-Cyhalothrin) is appropriate, but it would be helpful to briefly mention why these specific compounds were targeted.

2-The use of the modified QuEChERS method followed by GC/HPLC is standard and effective, though it would improve clarity if the reason for choosing both GC and HPLC (rather than one) is explained.

3. The abstract contain the reported validation parameters that are acceptable (e.g., R² > 0.99, recovery > 70%, RSD < 20%), but mentioning the actual number of pesticides tested would add value.

4. Some sentences are overly long and would benefit from simplification.

5-Minor grammatical issues exist, such as:

"method validation were tested" → should be "was tested".

"diverse group in okra matrix" → better phrased as "a diverse group of pesticide residues in the okra matrix".

6. The abstract does not clearly state what is novel or unique about this study compared to previous similar work. This should be addressed in one sentence.

7. Although the method is described as "simple, rapid, and cost-effective", no greenness assessment has been included to support its environmental friendliness. Given the increasing emphasis on green analytical chemistry, it is strongly recommended to evaluate the method’s greenness using one or more established tools such as the Analytical Eco-Scale, AGREE, or GAPI. This will enhance the relevance and sustainability value of the proposed method.

Reviewer #6: This MS developed an analytical method for measuring three pesticides in Okra. And this is a very comment method used for analyzing pesiticides, and there are well documented publications on this old topic, this study did not explain why they did this work and the current status of the analytical method for these pesticides.

overall, this study lacks of novelty.

**Do you want your identity to be public for this peer review?** For information about this choice, including consent withdrawal, please see our Privacy Policy

Reviewer #1: **Yes: ** Gaetan Glauser

Reviewer #2: No

Reviewer #3: **Yes: ** Peter Carl

Reviewer #4: No

Reviewer #5: **Yes: ** Rehab H Elattar

Reviewer #6: No

---

## [Author Response · Author response to Decision Letter 1]

20 Jun 2025

Clarification on queries paper entitled, “Method validation and Measurement uncertainty estimation of diverse group of Pesticide residues in Okra by GC/HPLC”

Journal requirements Queries Clarification

2. We noticed you have some minor occurrence of overlapping text with the following previous publication(s), which needs to be addressed

In your revision ensure you cite all your sources (including your own works), and quote or rephrase any duplicated text outside the methods section.

3. We note that you have provided funding information that is not currently declared in your Funding Statement. However, funding information should not appear in the Acknowledgments section or other areas of your manuscript.

4. We strongly recommend all authors decide on a data sharing plan before acceptance. 1. The ms. has been modified as per PLOS ONE’s style.

2. The overlapping text with the previous publication(s), on QC chart has been addressed.

All sources including my own have been cited and the duplicate text has been rephrased.

3. The funding-related text from the acknowledgement section of the manuscript has been deleted.

The funding statement has been mentioned in the cover letter.

4. All authors have accepted to allow the data sharing.

Reviewer no. Query Clarification

Reviewer no. 1

(i) With LOQs of about 0.25 mg/kg, these methods cannot reach the MRLs defined e.g. by the European Commission. The MRLs for okra as defined by Food Safety and Standards Authority India (FSSAI) for Contaminants, Toxins and Pesticide Residues, as per Regulations, 2011 are Thiamethoxam (0.5 mg/kg), Ethion (1mg/kg), and for lambda Cyhalothrin is 2mg/kg. Hence the pesticides whose MRLs in okra were found to be above the LOQ were selected for method validation.

(ii) There was no application to real samples to evaluate the applicability of the methods and the proportion of contaminated samples on a selection of plants. a) The study was performed to determine the extent of compliance of validation parameters as per SANTE guidelines and not monitoring of pesticide residues in okra. Thus the work on contaminated real-world okra samples has not been reported.

However, the validated method has been tested on real samples too and can be successfully applied to real samples as well.

(iii) Line 56 : « to control » Corrected

(iv) Line 58 : « may pose » Corrected

(v) Line 59 : « pesticide mixtures » Corrected

(vi) Line 69 : « precision » is mentioned twice Corrected

(vii) Line 93: “neonicotinoids Corrected

(viii) Line 113: make it clearer that you used an extraction protocol of yours which had been published earlier. Made clear statement regarding the extraction protocol

(ix) Figure 4: I cannot see the line for the observed values in the plot at 1 mg/kg. The corrected figure 4 of QC charts at LOQ of Ethion, lambda Cyhalothrin and Thiamethoxam have been incorporated in the figure file

(x) They are several grammatical and spelling mistakes in particular in the abstract and introduction. The grammatical and spelling mistakes have been corrected in the ms.

Reviewer no. 2

(i) Capitalization inconsistency in the title The inconsistency has been fixed as suggested

(ii) Abstract issues

a) The introduction lacks context on why pesticide detection in okra is significant.

Phrasing issues: e.g., “logical specificity” is unclear.

b) The abstract lacks clear context on the problem being addressed.

a)The novel aspect regarding method validation in okra alongwith significance of monitoring pesticide residues in okra have been addressed in the abstract.

b) The relevance of determining pesticide residues in public health and food regulation has also been mentioned.

(iii) Introduction issues

a) Redundant mentions of pesticide contamination and vague claims (e.g., "people apply pesticides mixture").

b) Redundant mention of ISO/IEC 17025 uncertainty requirements (lines 70–80).

c) Use of outdated or non-specific references (e.g., Pestology 1984 [2]).

d) Improve citation base with more recent references from 2015–2024. a) Since okra is cultivated during summer and rainy seasons which are conducive for insect infestation, farmers commonly apply mixture of insecticides for crop protection. Statement has been revised.

b) The sentence has been deleted

c) The reference has been removed

d) Recent references have been added.

(iv) Material and Methods issues

a) Lack of justification for pesticide selection beyond availability.

b) Inconsistent units (e.g., “mg/Kg” vs. “mg/kg”), overly condensed chromatography conditions

Validation parameters

c)The correlation coefficients (R²) and regression equations are present but lack residual analysis.

No comparative method or matrix challenge was mentioned.

Measurement Uncertainty

d)Clarify the meaning of “u1–u7” terms

e)Elaborate on why lower values validate reliability in vegetables.

a) Justification for pesticide selection with reference is given in Introduction

b) Necessary corrections regarding inconsistent units have been done in the ms. For the purpose of comparison overlaid chromatograms have been given in figure 2.

c) Residual analysis and comparison of standard methods for these pesticides in tomatoes and cucumbers has been done in Results and Discussion

d)The meaning of u1-u7 has been elaborated

e) Elaborated why lower values of MU validate reliability in vegetables

(v) Result and Discussion issues

a) The discussion does not critically compare GC vs. LC methods or address cross-matrix reproducibility.

b) References like “(15)” or “(Fig. 2 (i-iii))” are not adequately contextualized. The discussion lacks depth on method comparison or implications of the findings.

Figures and Tables

c) Figures referenced (e.g., Fig. 1, 2, 3, 4) are not embedded in the main text. This reduces comprehension and violates PLOS ONE’s formatting standards.

Conclusion

d)Summarizes the work adequately but contains redundancy with abstract and results

References

e) Duplicates found (Shrestha 2024 appears twice); fix citation style inconsistencies (some lack DOIs).

f)The citation style is inconsistent and lacks uniform formatting

a) Comparison of GC vs. LC methods or address cross-matrix reproducibility has been done in the text.

b) A comparative paragraph on how this method performs versus LC-MS/MS or published QuEChERS-GC methods has been added.

c) As per PLOS ONE’s style, Figure captions must be inserted in the text of the manuscript, immediately following the paragraph in which the figure is first cited so Figures were not embedded in the text but submitted as separate files.

d)Conclusion has been modified

e)The duplicacy of Shrestha 2024 reference has been removed. DOIs of all papers cited, has been given except the ones who’s DOIs are unavailable.

f)The citation style has been made consistent as per PLOS ONE author guidelines

(vi) Questions Inferred from Manuscript

1. Could this method be extended or adapted for detecting similar residues in other vegetables or high-water content matrices like cucumber?

2. How does this method compare detection limits and cost to advanced LC-MS/MS-based multi-residue protocols?

3. Are the measurement uncertainties consistent across different analysts or labs (interlaboratory validation potential)? T this

1.This method can be extended for detecting similar residues in other vegetables like bean, tomatoes, chili, cabbage, cauliflower, brinjal etc. where it has been tested. The method has not been validated on high-water content matrices like cucumber etc. but probably the QuEChERS method can be validated on high-water content matrices too as such studies have been reported in cucumber.

2. The detection limits of LC-MS/MS-based multi-residue protocols are definitely very low such that they can meet the EU MRLs but the LC-MS/MS is highly expensive and is not available in our lab.

Since we have performed analysis using HPLC and GC system we could detect up to the MRLs set by Food Safety and Standards Authority India (FSSAI) for Contaminants, Toxins and Pesticide Residues, as per Regulations, 2011. The method validation parameters as per SANCO guidelines could be achieved in the process.

3. The measurement uncertainties are based upon intralaboratory validation

Reviewer 3

(i) a) Lack of Supporting Data and Transparency in Calculation

A critical shortcoming of the manuscript is the absence of supporting data. Essential information and raw data underlying the method validation and uncertainty estimation are not provided.

The manuscript does not explain how the measurement uncertainty (MU) was calculated, nor are the input values or assumptions documented. For a method that may be used in ISO/IEC 17025-accredited laboratories, this level of detail is imperative.

A supplementary section containing raw data, calculation spreadsheets, and validation outputs would greatly enhance transparency and reproducibility.

b) Lack of Real Sample Analysis and Reference Materials

The study appears to rely solely on spiked matrixes without including contaminated real-world okra samples. This limits the applicability and relevance of the method to actual monitoring scenarios.

c) Matrix Considerations and Source Variability

The authors did not account for potential variability introduced by different varieties or sources of okra. Since agricultural matrices can vary significantly, especially in complex plant matrices like okra, it is essential to consider source-related effects on method performance.

d) Selectivity and Interferences

The manuscript lacks a proper assessment of selectivity. In the absence of mass spectrometric detection, structural analogs or co-eluting compounds may compromise the method's specificity. E. g, there is no mention of interference studies involving structurally similar pesticides that may be present in real samples.

e) e)Diastereomer Considerations for Lambda-Cyhalothrin

f) Lambda-Cyhalothrin is known to be a mixture of diastereomers that are separable by GC. The paper does not address whether these isomers were resolved in the method, and if not, how quantification was handled. This is a critical omission, as the relative proportions of isomers can influence the accuracy of the method.

g) f) Inadequate Method Description and Reproducibility Concerns

The QuEChERS extraction protocol is not described in sufficient detail to ensure reproducibility. The manuscript refers to a paper that references another paper, instead of clearly describing the procedure used. A self-contained, step-by-step description is necessary for validation and replication.

h) g) Robustness and Sample Preparation

An insufficient robustness study was conducted, that relied solemnly on the instrumental (detection) parameters. Key aspects of sample preparation (e.g., drying, grinding, extraction, cleanup) are major contributors to variability and measurement uncertainty. The impact of these factors should have been assessed both in the robustness study and in the uncertainty estimation.

i)

Minor Comments

1. Figures and Visual Data Quality

Figure 1: The chemical structures lack uniformity in style; the structures of Ethion and Cyhalothrin appear distorted.

Figure 2B: The observed value line is missing, making it impossible to interpret the graph properly.

2. Units

The authors should use SI-compliant units such as “mg kg⁻¹” or “mg/kg,” following recommendations by IUPAC, BIPM, and other standards bodies.

3. Language and Typos

The manuscript requires language polishing. Several typographical and grammatical errors are present. For example:

Line 131: “Europeon [sic!] Commission” should be “European Commission”

Line 69: The phrase “assays of precision, bias, linearity, precision…” is redundant and unclear. Saa a) Raw Data will be provided in the supplementary section for transparency in calculation

Raw Data indicating calculation of measurement

uncertainty (MU) will also be provided in the

supplementary section

b)The study was performed to determine the extent of compliance of validation parameters as per SANTE guidelines and not monitoring of pesticide residues in okra. The validated method has been tested on real samples too.

c) The method validation study was undertaken on the main variety of okra (Abelmoschus esculentus (L.) which is cultivated in this region.

Though significant variation is observed in various morphological traits like plant height, pod length, number of branches, and yield in different varieties of okra but they are not known to affect the method validation parameters.

d) Due to non availability of mass spectrometric detection, pesticides were analysed singly so that there is no interference of structurally similar pesticides. Moreover, the okra vegetable taken for method validation was from the vegetable research centre (VRC) plot where no pesticide was sprayed.

e) Lambda-Cyhalothrin was validated as such in the form of a single compound throughout the study. Resolution of isomers was not observed.

f) The reference of the detail QuEChERS extraction protocol was already given in the text. However, a step-by-step description which is necessary for validation and replication has been added in the methodology section.

g) Robustness was conducted on the instrumental parameters like change in flow rate, method programming, change in mobile phase concentration and detection wavelength, which are mainly responsible for determining the robustness of the validated method. Similar type of studies regarding robustness of the method have been reported in other method validation studies too.

1. Figure 1 The chemical structures of Ethion and Cyhalothrin have been corrected

Figure 2B The observed value line is corrected and the graph has been interpreted properly

2. Correction regarding use of units has been done

3. Typographical and grammatical errors have been corrected

Line 131: “Corrected as suggested

Line 69: The phrase “assays of precision, bias, linearity, precision has been clarified.

Reviewer 4 1) 1) At abstract, line number 36 please modify 0.3mg/kg to be 0.30 mg/kg.

2) Please justify the alignment for the whole manuscript.

3) Please try to increase the figures resolution

4) At Introduction line 56: modify from Several groups of pesticides are used to for controlling weeds to be Several groups of pesticides are used for controlling weeds. 1) Done as suggested

2) The alignment of the ms. has been changed to justified.

3) The figure resolution has been increased

4) Done as suggested and modified as per Reviewer 1

Reviewer 5

1-The selection of pesticides (Thiamethoxam, Ethion, and lambda-Cyhalothrin) is appropriate, but it would be helpful to briefly mention why these specific compounds were targeted.

2-The use of the modified QuEChERS method followed by GC/HPLC is standard and effective, though it would improve clarity if the reason for choosing both GC and HPLC (rather than one) is explained.

3.The abstract contain the reported validation parameters that are acceptable (e.g., R² > 0.99, recovery > 70%, RSD < 20%), but mentioning the actual number of pesticides tested would add value.

4. Some sentences are overly long and would benefit from simplification.

5.Minor grammatical issues exist, such as:

"method validation were tested" → should be "was tested".

"diverse group in okra matrix" → better phrased as "a diverse group of pesticide residues in the okra matrix".

6.The abstract does not clearly state what is novel or unique about this study compared to previous similar work. This should be addressed in one sentence.

7. Although the method is described as "simple, rapid, and cost-effective", no greenness assessment has been included to support its environmental friendliness. Given the increasing emphasis on green analytical chemistry, it is strongly recommended to evaluate the method’s greenness using one or more established tools such as the Analytical Eco-Scale, AGREE, or GAPI. This will enhance the relevance and sustainability value of the proposed method.

1. The

---

## [Decision Letter · Decision Letter 1]

7 Jul 2025

Dear Dr. Srivastava, 

Thank you for submitting your manuscript to PLOS ONE. After careful consideration, we feel that it has merit but does not fully meet PLOS ONE’s publication criteria as it currently stands. Therefore, we invite you to submit a revised version of the manuscript that addresses the points raised during the review process.

We look forward to receiving your revised manuscript.

Kind regards,

Trung Quang Nguyen

Academic Editor

PLOS ONE

Journal Requirements:

Reviewers' comments:

Reviewer's Responses to Questions

**Comments to the Author**

Reviewer #1: (No Response)

Reviewer #4: All comments have been addressed

Reviewer #5: All comments have been addressed

2. Is the manuscript technically sound, and do the data support the conclusions?

Reviewer #1: Yes

Reviewer #4: Yes

Reviewer #5: Yes

3. Has the statistical analysis been performed appropriately and rigorously?

Reviewer #1: Yes

Reviewer #4: Yes

Reviewer #5: Yes

4. Have the authors made all data underlying the findings in their manuscript fully available?

Reviewer #1: Yes

Reviewer #4: Yes

Reviewer #5: Yes

5. Is the manuscript presented in an intelligible fashion and written in standard English?

Reviewer #1: Yes

Reviewer #4: Yes

Reviewer #5: Yes

Reviewer #1: The authors have addressed my comments relatively adequatetly. Yet, I am still a little frustrated not to see any data on real samples, in particular since the authors mentioned that they have some. Unless they have strong arguments not to publish these data in the present paper, I strongly recommend they do so. Even if the data are intended for another paper, a small subset could be published along with this methodological paper to illustrate the applicability of the method and reinforce the paper.

Reviewer #4: Reviewer comments:

I sincerely appreciate the editor's suggestion that I revise this manuscript.

After the second revision, I suggest that the manuscript can be accepted in its current form.

Reviewer #5: The authors have revised and answered all the questions. The current revised manuscript can be published in PLOS One journal.

**Do you want your identity to be public for this peer review?** For information about this choice, including consent withdrawal, please see our Privacy Policy

Reviewer #1: **Yes: ** Gaetan Glauser

Reviewer #4: No

Reviewer #5: **Yes: ** Rehab Hamdy Elattar

---

## [Author Response · Author response to Decision Letter 2]

18 Jul 2025

Response to the academic editor and reviewers on paper entitled, “Method validation and Measurement uncertainty estimation of diverse group of Pesticide residues in Okra by GC/HPLC”

S. No Comments to the Author Response from reviewer Clarification

1. If the authors have adequately addressed your comments raised in a previous round of review and you feel that this manuscript is now acceptable for publication, you may indicate that here to bypass the “Comments to the Author” section, enter your conflict of interest statement in the “Confidential to Editor” section, and submit your "Accept" recommendation.

Reviewer no.# 1 No comments No clarification required

Reviewer no.# 4 All comments have been addressed No clarification required

Reviewer no.# 5 All comments have been addressed No clarification required

2. Is the manuscript technically sound, and do the data support the conclusions?

Reviewer no.# 1 Yes No clarification required

Reviewer no.# 4 Yes No clarification required

Reviewer no.# 5 Yes No clarification required

3. Has the statistical analysis been performed appropriately and rigorously?

Reviewer no.# 1 Yes No clarification required

Reviewer no.# 4 Yes No clarification required

Reviewer no.# 5 Yes No clarification required

4. Have the authors made all data underlying the findings in their manuscript fully available?

Reviewer no.# 1 Yes No clarification required

Reviewer no.# 4 Yes No clarification required

Reviewer no.# 5 Yes No clarification required

5. Is the manuscript presented in an intelligible fashion and written in standard English?

Reviewer no.# 1 Yes No clarification required

Reviewer no.# 4 Yes No clarification required

Reviewer no.# 5 Yes No clarification required

6. Review Comments to the Author

Reviewer no.# 1 The authors have addressed my comments relatively adequatetly. Yet, I am still a little frustrated not to see any data on real samples, in particular since the authors mentioned that they have some. Unless they have strong arguments not to publish these data in the present paper, I strongly recommend they do so. Even if the data are intended for another paper, a small subset could be published along with this methodological paper to illustrate the applicability of the method and reinforce the paper.

Clarification :The real samples of okra are laced with a number of different pesticides which could interfere while method validation of these pesticides. Since the present study was specifically conducted to validate a method for estimation of the three commonly applied pesticides the data on real samples was not much generated.

Reviewer no.# 4 I sincerely appreciate the editor's suggestion that I revise this manuscript.After the second revision, I suggest that the manuscript can be accepted in its current form.

Clarification has been given and an explanation regarding the query has also been given in the revised ms.

Reviewer no.# 5 The authors have revised and answered all the questions. The current revised manuscript can be published in PLOS One journal. No clarification required

7. PLOS authors have the option to publish the peer review history of their article (what does this mean?). If published, this will include your full peer review and any attached files. Reviewer no.# 1 Yes -

Reviewer no.# 4 No -

Reviewer no.# 5 Yes -

---

## [Decision Letter · Decision Letter 2]

6 Aug 2025

Method validation and Measurement Uncertainty Estimation of Pesticide Residues in Okra by GC/HPLC

PONE-D-25-17690R2

Dear Dr. Anjana Srivastava,

We’re pleased to inform you that your manuscript has been judged scientifically suitable for publication and will be formally accepted for publication once it meets all outstanding technical requirements.

Kind regards,

Trung Quang Nguyen

Academic Editor

PLOS ONE

Additional Editor Comments (optional):

Reviewers' comments:

Reviewer's Responses to Questions

**Comments to the Author**

Reviewer #1: All comments have been addressed

2. Is the manuscript technically sound, and do the data support the conclusions?

Reviewer #1: Yes

3. Has the statistical analysis been performed appropriately and rigorously?

Reviewer #1: Yes

4. Have the authors made all data underlying the findings in their manuscript fully available?

Reviewer #1: Yes

5. Is the manuscript presented in an intelligible fashion and written in standard English?

Reviewer #1: Yes

Reviewer #1: (No Response)

**Do you want your identity to be public for this peer review?** For information about this choice, including consent withdrawal, please see our Privacy Policy

Reviewer #1: **Yes: ** Gaetan Glauser

---

## [Editor Report · Acceptance letter]

PONE-D-25-17690R2

PLOS ONE

Dear Dr. Srivastava,

I'm pleased to inform you that your manuscript has been deemed suitable for publication in PLOS ONE. Congratulations! Your manuscript is now being handed over to our production team.

Kind regards,

on behalf of

Dr. Trung Quang Nguyen

Academic Editor

PLOS ONE